nanotechnology

biosensors, gold nanostars, glucose oxidase, plasmonic, synthesis

**Author for correspondence:**
Masauso Moses Phiri
e-mail: missiphiri@gmail.com

This article has been edited by the Royal Society of Chemistry, including the commissioning, peer review process and editorial aspects up to the point of acceptance.

# Seedless gold nanostars with seed-like advantages for biosensing applications

Masauso Moses Phiri, Danielle Wingrove Mulder and Barend Christiaan Vorster

Centre for Human Metabolomics, North-West University, P/Bag X6001, Hoffman Street, Potchefstroom 2520, South Africa

MMP, 0000-0001-7653-7988; DWM, 0000-0002-6970-7392; BCV, 0000-0003-2371-288X

Gold nanostars (AuNSs) are seen as promising building blocks for biosensors with potential for easy readouts based on naked-eye and ultraviolet–visible spectroscopy detection. We present a seedless synthesis strategy for AuNSs that has the advantages of the seeded methods. The method used ascorbic acid as a reducing agent and silver nitrate as an anisotropic growth control assisting agent. AuNSs with multiple branches and a diameter of 59 nm were produced. They showed good stability when capped with PVP and modified with an enzyme in relatively strong ionic conditions. We investigated their application in plasmonic sensing by modifying them with glucose oxidase and detection of glucose. The AuNSs were found to be a good scaffold for the enzyme, proved to be stable and sensitive as transducers. Thus, the AuNSs showed good promise for further applications in plasmonic biosensing for *in vivo* biomedical diagnosis.

## 1. Introduction

Significant advances have been made in the synthesis of gold nanoparticles (AuNPs) to an extent that other nanoshapes are readily obtained using appropriate templating agents. Different shapes, such as nanorods, nanocubes, nanoprisms, nanowires, nanoboxes, nanoshells, triangular, hexagonal shapes and even nanostars, have been produced [1,2]. Among the various geometries of gold nanoparticles, gold nanostars (AuNSs), alternatively referred to as nanoflowers [3], nanourchins or multi-branched gold nanoparticles, have received much attention in recent years. This is because of their catalytic activity, molecular detection and biological applications in immunoassays, dark field imaging of cells and in plasmonic biosensors [4–7]. AuNSs have been capable building blocks for near-infrared (NIR) absorption and surface-enhanced Raman scattering applications

because of their many branches and sharp tips. Owing to their high-aspect-ratio spikes that localize the low-energy plasmon mode at their tips, AuNSs give a dominant localized surface plasmon resonance (LSPR) peak in the NIR region. AuNSs are thus an attractive platform for LSPR biosensing application for diagnostics purposes with potential for easy readouts based on colorimetric and ultraviolet–visible (UV–vis) spectroscopy detection [8–11].

The last decade has seen advancements in the various synthesis methods for AuNSs. A rough general classification of the various synthesis strategies falls into two main categories: the seeded-growth and non-seeded-growth methods [12]. The seeded-growth technique is a popular method for the synthesis of monodispersed gold nanostars. In this method, pre-synthesized seeds (AuNPs) are used as nucleation points where additional material is deposited for growth of the branches [13,14]. The synthesis process is involved in the reduction of hydrochloroauric acid (HAuCl$_4$) with ascorbic acid— or other reducing agents—on preformed gold seeds in the presence of a surfactant at room temperature [14]. Addition of silver nitrate (AgNO$_3$) at different growth process stages of the nanocrystals increases the degree of control of the nanostar shape produced [4]. However, this method has some setbacks, one of which is the complication caused by the various stabilizing agents and surfactant in the post-synthesis cleaning of the nanostars [14]. Recently, a surfactant- and polymer-free shape control synthesis method was proposed that was enabled by a unified theoretical framework of nanocrystal synthesis. Nanostars, among other morphologies, were synthesized with this simple green-chemistry method for catalysis and surface-enhanced Raman scattering [15].

Recent advances in seedless strategies have seen the use of 'green' chemicals, such as N-2-hydroxyethylpiperazine-N-2-ethanesulfonic acid (HEPES) as reducing and stabilizing agent. In this one-pot synthesis strategy, nuclei evolve to form nanocrystal seeds which get to be bigger particles through the direct addition of metal atoms. The presence of piperazine in HEPES is thought to be responsible for branch formation on the nanocrystals [12,16,17]. Compared to the seeded-growth strategy, this technique has fewer complications, with the advantage of being completed in one single step and pot. Some protocols are carried out without using surfactants, making the post-synthesis purification of the AuNSs formed less problematic. However, it has a number of disadvantages, one of which is its inability to control the dimensions of the resulting nanostars leading to polydispersity in the shapes and sizes of particles produced. Another notable setback is the high sensitivity to changes in the reaction parameters such as concentrations of reagents, pH and temperature, which affects the growth process and reproducibility of the nanocrystals [4,11,13,14,16].

In an attempt to optimize the HEPES-mediated method to yield more monodispersed AuNSs, an in-house method was developed recently in which a specific amount of AgNO$_3$ was added to aid in the growth process of the nanostar branches. It was reported to have yielded less polydispersed AuNSs compared to the earlier reported method [18]. However, this method is relatively much slower in comparison to the seeded methods reported. It takes over 30 min for AuNSs synthesis to be completed. AuNSs produced were smaller compared to the seeded ones, thereby having an LSPR peak (which is size dependent) at less than 630 nm, making them minimally NIR sensitive. Other setbacks typical of non-seeded synthesis also apply to it, such as high sensitivity to changes in the conditions and concentrations of precursor reagents, such as pH of the HEPES buffer, temperature and HAuCl$_4$ concentrations, strongly affects the reproducibility of the synthesized AuNSs [4,11,14,16,19].

Herein, a seedless one-pot method is reported that leveraged the advantages of the seeded strategies and those of seedless ones. Advantages of seeded methods, such as rapidity, monodispersity, controlled growth of branches, bigger sizes of nanostars produced for greater LSPR sensitivity, as well as the simplicity and non-harmful reagents of the seedless method, were combined in this method. In this procedure, gold nuclei were reduced with ascorbic acid. AgNO$_3$ was added during synthesis to control the growth of the nanostar branches with precision. The whole synthesis procedure took approximately 2 min to be completed. The stability of the synthesized stars in solution of ionic strength and varying pH was investigated. Furthermore, they were also investigated for plasmonic colorimetric sensing using glucose measurement as a model.

# 2. Material and methods

## 2.1. Materials and instrumentation

Hydrochloroauric acid (HAuCl$_4$), glucose oxidase (GOx), trisodium citrate, silver nitrate (AgNO$_3$), ascorbic acid, sodium chloride (NaCl), polyvinylpyrrolidone (PVP) (molecular weight 10 000),

hydrochloric acid (HCl), glucose, 3 3′-dithiobis (sulfosuccinimidyl propionate) (DTSSP), 2-(N-morpholino)ethanesulfonic acid (MES) at pH 6, (4-(2-hydroxyethyl)-1-piperazineethanesulfonic acid) acid (HEPES) at pH 7.4, Tris-acetate-EDTA (TAE) at pH 8, bovine serum albumin (BSA), ethanolamine and phosphate buffered saline (PBS) at pH 7.4 were all purchased from Sigma-Aldrich, South Africa. All glassware was stripped with Aqua Regia prior to use for synthesis. Ultrapure water (ddH$_2$O) was pre-prepared with a Milli-Q ultra-pure system (18.2 MΩ cm$^{-1}$).

UV−vis spectroscopy analysis was carried out by spectral scanning (300−900 nm) on a HT Synergy (BioTEK) microplate reader. The transmission electron microscopy (TEM) and energy-dispersive X-ray spectroscopy (EDS) analyses were performed on a Tecnai F20 high transmission electron microscope (HR-TEM) at an accelerating voltage of 200 kV. Samples for TEM were prepared by applying 20 μl of nanoparticle suspension onto carbon 200 mesh copper grids (Agar Scientific), followed by drying overnight prior to imaging. The particle-size distribution was estimated by measuring the size of approximately 100 nanoparticles in different grid regions. The EDS analysis was normalized to remove carbon and copper from the total chart as the samples were fixed on carbon-coated copper grids for analysis. Dynamic light scattering (DLS) was used to estimate the hydrodynamic diameter of the AuNSs. It was performed on a Zetasizer Nano (Malvern) in backscatter mode using Zetasizer v. 6.20 software in capped polystyrene cuvettes.

Agarose gel electrophoresis of the AuNSs functionalized with PVP with and without GOx was carried out using a Baygene, BG-power, Vacutec electrophoresis gel apparatus. Agarose gel (0.75%) was prepared by weighing 0.375 g of agarose and dissolving it in 50 ml of TAE in the microwave. The gel was poured into the casting tray and allowed to set. Small aliquots (30 μl; a mixture of 2 parts AuNSs and 1 part glycerol (80%)) of the samples were loaded onto the gel and run at 40 volts for 45 min. Gel images were captured and transferred to the computer.

## 2.2. Gold nanostars synthesis

All experiments were carried out at room temperature unless otherwise stated.

Seedless silver and ascorbic acid-assisted nanostars (*seedless-AuNSs*) were synthesized as follows: 10 ml of ddH$_2$O was acidified with 10 μl of 1 M HCl. Following this, 50 μl of 100 mM ascorbic acid was added under mild stirring and 50 μl of 50 mM HAuCl$_4$ was then added to the mixture. Immediately after this, 50 μl of 10 mM AgNO$_3$ was added to the solution which resulted in a deep blue colour change within a few seconds. Finally, 500 μl of 2.5 mM PVP was added, after which the solution was centrifuged for 90 min at 4000$g$. The pellet was then recovered and resuspended in 1 ml of ddH$_2$0.

For purposes of comparison, two other syntheses of AuNSs were done; the one via a seeded method (*seeded-AuNSs*), and the other via the in-house HEPES-mediated method (*HEPES-AuNSs*). The seeded-AuNSs were synthesized following previously published methods with minor modifications [20,21]. The seeds were synthesized according to the Turkevich−citrate reduction method [22]. Briefly, 0.25 mM HAuCl$_4$ was added to 50 ml ddH$_2$O and heated to 95°C. Thereafter, 1300 μl 1% trisodium citrate was immediately added to the mixture resulting in seed formation. The seeds were then stored at 4°C until usage. AuNSs were synthesized by acidifying 10 ml of 0.25 mM HAuCl$_4$ with 10 μl 1 M HCl followed by the addition of 5 μl of the seed solution. Thereafter, 50 μl 10 mM AgNO$_3$ and 50 μl 100 mM ascorbic acid were added simultaneously to the solution under mild stirring. Colour change was observed (from pale yellow to deep blue). The nanostars were coated with 350 μl 2.5 mM PVP. Sample clean-up was done by centrifugation at 3000$g$ for 2 h and resuspended in 1 ml ddH$_2$0.

The HEPES-AuNSs were synthesized according to an in-house HEPES-modified method for seedless nanostars [18]. In a typical synthesis, 2 ml of 100 mM HEPES was mixed with 3 ml ddH$_2$O in a 5 ml tube. This was followed by the addition of 20 μl 50 mM HAuCl$_4$ and 4 μl 1 mM AgNO$_3$. The capped tube was inverted a few times to mix. The solution was left to stand for approximately 30 min, in which the solution's colour changed from light yellow to slightly purple and finally a deep blue.

## 2.3. Modification of AuNSs with GOx

The seedless AuNSs were chosen for GOx modification and further investigations. The method suggested by Filbrun *et al.* with minor modifications was followed to attach GOx to the AuNSs with DTSSP [23]. Briefly, AuNSs was suspended in 2 ml PBS to which 100 μl 5 mM DTSSP was added to the solution. After 3 h incubation, the sample was centrifuged at 3000$g$ for 1 h to remove excess DTSSP. The sample was then resuspended in 2 ml PBS after which 250 μl of 5 mg ml$^{-1}$ GOx was added and left to react for 2 h. Then 100 μl of BSA 1 mg ml$^{-1}$ and 100 μl of ethanolamine 10 mM

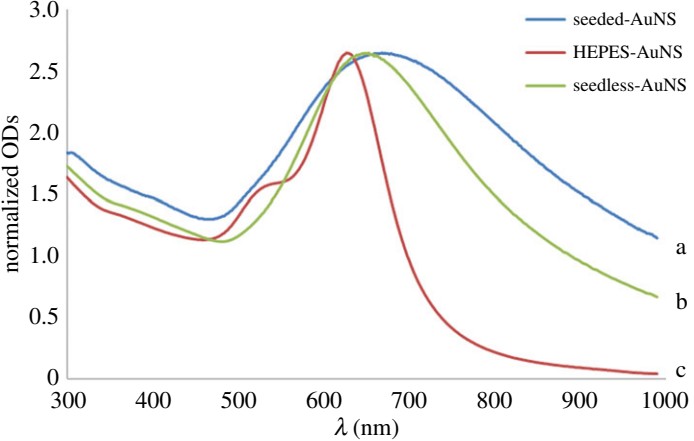

**Figure 1.** Comparison of UV–vis spectra of seeded-AuNSs (a), seedless-AuNSs (b) and HEPES-AuNSs (c).

were added as blockers and left to react for an extra hour. The solution was centrifuged and the pellet resuspended in 1 ml MES buffer. The GOx-modified seedless-AuNSs were characterized on a UV–vis spectrophotometer. DLS was used to determine the sizes of the control PVP-coated and GOx-modified seedless AuNSs. Gel electrophoresis was done to determine the sizes and charges of these AuNSs.

## 2.4. Colloidal stability

The stability of the GOx-modified seedless AuNSs was investigated in solutions of high ionic strength. The gold nanostars were centrifuged and resuspended in 300 mM NaCl and the UV–vis spectra were obtained after 2 h incubation.

## 2.5. Feasibility of plasmonic sensing using glucose measurement

The GOx-modified seedless AuNSs were assessed in their ability to be used as plasmonic nanosensors. The mechanism of sensing was the enzyme-guided enlargement of AuNSs through the production of hydrogen peroxide ($H_2O_2$) from glucose breakdown. First, 15 µl of 100 mM glucose was added to GOx-modified seedless-AuNSs in 10 mM MES buffer and incubated at 37°C for 1 h. Then 15 µl of a 0.1 mM $AgNO_3$ and 40 mM $NH_3$ solution were added to the mixture (with a total volume of 200 µl) to trigger the reduction of the silver ions ($Ag^+$) on the AuNSs. UV–vis spectral changes were measured immediately. Furthermore, a range of glucose concentrations were added to the assay solution containing the GOx-modified seedless AuNSs to investigate their possibility as signal transducers to differentiate between varying concentrations of an analyte in a diagnostic assay.

# 3. Results and discussion

## 3.1. Characterization of the gold nanostars

Figure 1 displays a comparison of the UV–vis spectra of the AuNSs synthesized with the three different synthesis methods. The spectra observed had typical LSPRs of star-shaped nanoparticles as judged by the longer wavelengths and the broad peaks. Increased aspect ratios of the branches make the longitudinal components of the plasmon band become more intense and red-shifted relative to the LSPR of spherical particles [5,10]. The maximum absorption of the seeded-AuNSs was at about 668 nm, while the seedless-AuNSs and HEPES-AuNSs were absorbed at slightly shorter wavelengths of 653 nm and 627 nm, respectively. Particle sizes and aspect ratios of the spikes are known to govern such optical properties [10,11] as observed of these nanostars.

TEM analysis (figure 2) of the synthesized nanostars showed that the seeded-AuNSs (a) were slightly larger at a diameter of 60 ± 5 nm compared to the seedless-AuNSs (b) and HEPES-AuNSs (c) with diameters of 59 ± 5 and 44 ± 4 nm, respectively. The seeded-AuNSs and seedless-AuNSs had sharper tips of increased lengths compared to the HEPES-AuNSs. The newly synthesized seedless-AuNSs were comparable in morphology and size relative to the seeded-AuNSs.

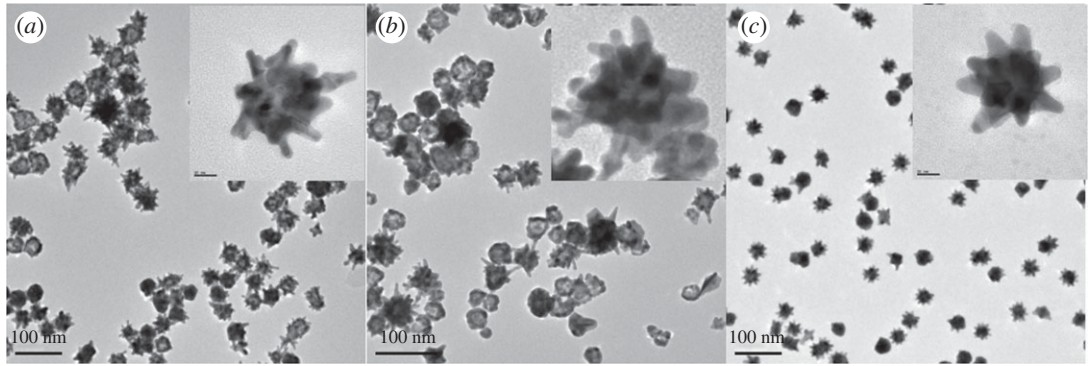

**Figure 2.** TEM images of seeded-AuNSs (*a*), seedless-AuNSs (*b*) and HEPES-AuNSs (*c*). The insets show the magnified TEM images of the respective AuNSs at 10 nm.

The HEPES-AuNSs were more uniformly formed with greater size control compared to both the seeded- and seedless-AuNSs. The representative TEM images of the all the three types of AuNSs synthesized showed that all had a high yield of branched particles with the presence of a few spherical morphologies.

This new synthesis method was discovered by strategically changing the order of addition of required reagents and eliminating the gold colloidal seeds as used in the original method by Yuan *et al.* [21]. The seeded method followed the procedure of first adding colloidal seeds and HAuCl$_4$ to the solution followed by ascorbic acid and AgNO$_3$ which are added simultaneously [21]. But, in this new proposed method, ascorbic acid was instead added first soon after lowering the pH of ddH$_2$O with HCl. Only then was HAuCl$_4$ added to the mixture, and AgNO$_3$ immediately (10 s) after. As HAuCl$_4$ was in the process of being reduced by ascorbic acid, AgNO$_3$ was added for nanostar formation, without which nanorods and/or nanospheres are formed [21].

The EDS analysis showed that these stars had almost 100% Au with no Ag showing up. Thus, it is thought that the main role of silver is to aid the growth of branches on the Au on certain crystallographic facets [20,21]. Addition of a small amount of HCl is thought to promote further red-shifting in the LSPR of the nanostars. It was observed that even when HCl was not added in the synthesis procedure, gold nanostars were still formed. Yuan *et al.* reported that the localized surfaced plasmon resonance of nanostars synthesized at higher pHs such as 7 and 10 was more blue-shifted compared to those synthesized with HCl at pHs 3 and 4 [21]. They further investigated the effect of lowering the pH by substituting HCl with nitric acid. They observed that the nanostars were not formed at all. This led to the postulation that the Cl-ions could be involved in the formation of the gold nanostars [21]. However, the effects of addition or omission of HCl on the size of the nanostars were not investigated in this study.

This new method deals with the problem that was reported by Yuan *et al.* [21] in their attempt to synthesize nanostars without seeds. They report yielding nanostars of diameters greater than 100 nm [21], compared to 59 nm nanostars in this method. This new method also removed the need for careful addition of AgNO$_3$ and ascorbic acid at the same time and fast. In the seeded method reported, if the AgNO$_3$ was added too early, no nanostars are formed due to the precipitation of silver chloride. If the AgNO$_3$ was added too late, the HAuCl$_4$ would have already been reduced to larger gold nanospheres and nanorods. But in this new method, there is a stepwise addition of the reagents. The time for the addition of AgNO$_3$ was observed to be very important in this method. In the experimental protocol, AgNO$_3$ was added about 10 s after the addition of HAuCl$_4$. This led to star formation. But, prolonged delay of over 30 s in the addition of AgNO$_3$ led to the formation of other nanoshapes [21,24,25]. Without ascorbic acid, HAuCl$_4$ was not reduced to form any nanostructures. Other studies using the ascorbic acid silver-assisted nanostar recipes adequately report the experimental effects of the various combinations of ascorbic acid in relation to the HAuCl$_4$, as well as adding varying quantities of silver nitrate to the mixture [21,24,25]. With regard to time factors in the synthesis methods, the seedless-AuNSs took less than 2 min for a complete synthesis, whereas the HEPES-mediated ones took about 30 min. For the seeded ones, the actual star synthesis is comparable in time to the seedless ones. But when the time for the spherical gold nanoparticle synthesis and maturation is factored in, it took the longest time to be completed of the three strategies.

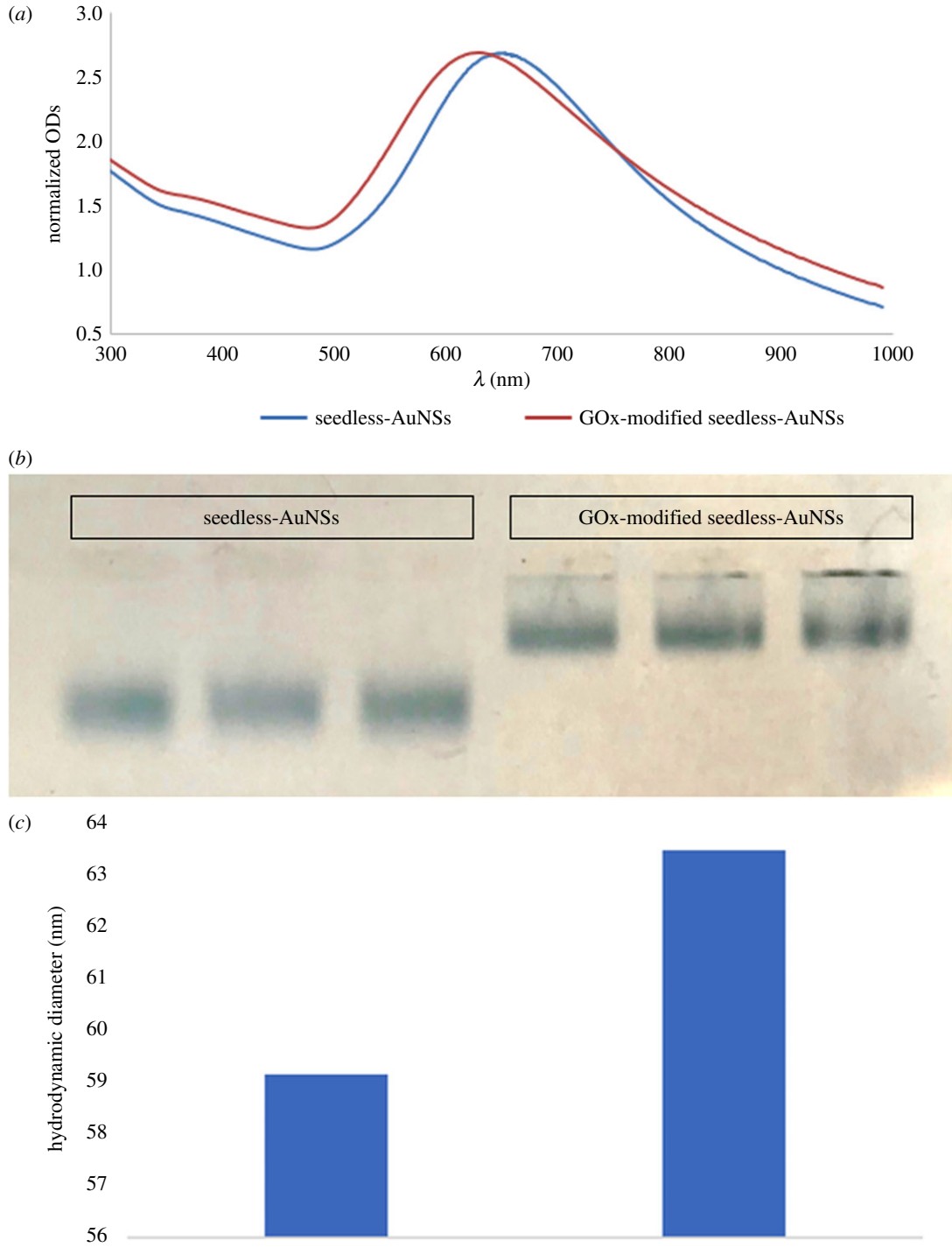

**Figure 3.** (a) Normalized UV–vis spectra of the control seedless-AuNSs and GOx-modified seedless-AuNSs. (b) Agarose gel electrophoresis of the control nanostars and those modified with GOx. (c) The hydrodynamic diameters of the nanostars before and after modification with GOx.

## 3.2. Modification of AuNSs with glucose oxidase

To test the application of seedless-AuNSs as nanosensors, they were covalently modified with GOx using DTSSP (figure 3). The UV spectrum of the seedless-AuNSs modified with $6.25 \times 10^{-4}$ g ml$^{-1}$ GOx was observed to blue-shift relative to control PVP-coated seedless-AuNSs (3–I). The maximum OD of the GOx-modified seedless-AuNSs shifted from 653 to 631 nm (by 22 nm). The shift was not accompanied by any broadening of the peak indicating the non-aggregation of the AuNSs at this point. Gel

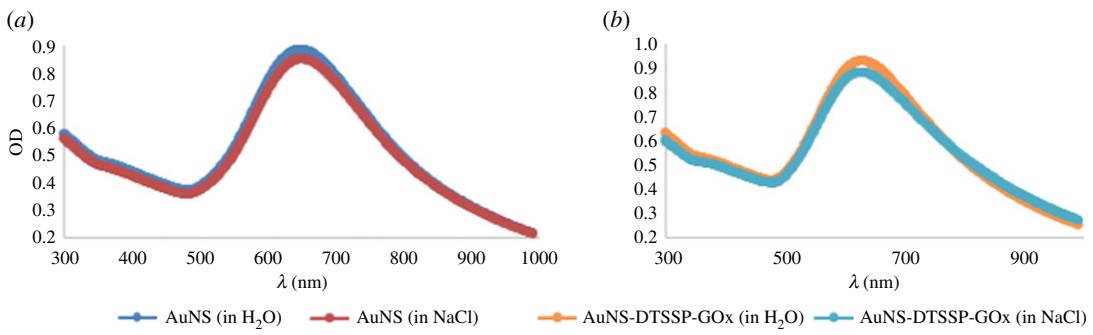

**Figure 4.** Images showing the stability of the stars in salt (*a*) for control seedless-AuNSs and (*b*) for GOx-modified seedless-AuNSs.

| | | | | | | |
|---|---|---|---|---|---|---|
| AuNS-GOx | √ | | √ | √ | √ | √ |
| glucose | | √ | | √ | √ | √ |
| Ag⁺ | | √ | √ | | √ | √ |
| base | | √ | √ | √ | | √ |

**Figure 5.** Feasibility of glucose sensing with GOx-modified nanosensors. Glucose: 2.5 mM, Ag$^+$: 0.1 mM and pH: >9.

electrophoresis was used to qualitatively confirm the binding of the enzyme (a: control PVP-AuNSs and b: GOx-AuNSs) to the gold nanostars (figure 3*b*). The gel migration decreased with the GOx-modified seedless-AuNSs compared to the control AuNSs with PVP coating only. DLS analysis (figure 3*c*) confirmed the quantitative growth in hydrodynamic size of the particles functionalized with GOx. These results put together indicate that the LSPR characteristics, size and surface charge of the particles have changed suggesting the attachment of the AuNSs with GOx in a qualitative manner.

Application of protein-modified gold nanostars in physiological pH and ionic environment requires that they are stable in solutions containing high concentration of proteins and salts. Thus, the stability of the GOx-modified seedless AuNSs was tested and proved in 0.3 M NaCl and pH 7 [10,26]. The protein-modified nanosensors showed great stability with no observed variation of the LSPR (figure 4). There was no reduction in absorption or broadening of the peak, indicating the integrity of the nanostructures in this environment. This is particularly important because gold nanoparticles can aggregate in solutions with high ionic strength when their surfaces are not adequately coated, which will lead to broadening of the peaks and shifts in LSPRs, thereby affecting the reliability of their detections [10].

## 3.3. Feasibility of glucose sensing

The feasibility of plasmonic colorimetric sensing, using the synthesized seedless-AuNSs, was investigated using glucose as a model analyte. Various control experiments were carried out, as shown in figure 5. No appreciable colour changes were observed in the absence of AuNSs, glucose, AgNO₃ or base. After an hour's incubation at 40°C, the only solution that showed significant colour change, as well as a large LSPR shift, was the one that had all the necessary components in it. This observation could be attributed to the enzyme-guided growth of AuNSs, yielding change in colour. The AuNS growth was induced by the coating of reduced silver on the surface as seeding points leading to epitaxial growth. Prior to this, the silver was reduced by $H_2O_2$ generated from glucose oxidation by GOx at pH 9. This experiment showed the absolute necessity of all the components to have a significant plasmonic shift and colour change.

Scheme 1 represents the mechanistic aspect of the plasmonic colorimetric strategy for glucose sensing observed in the feasibility experiment.

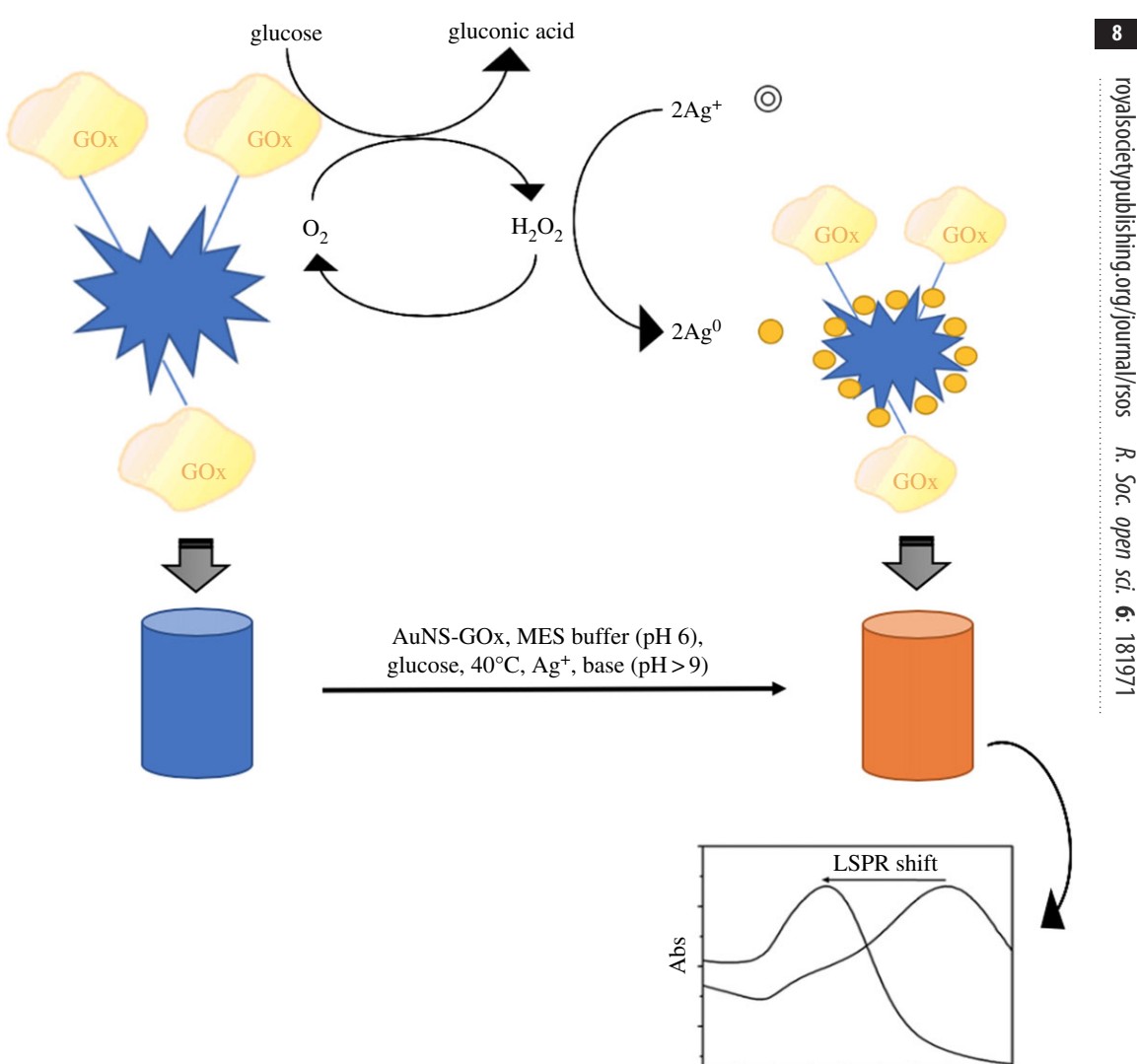

**Scheme 1.** Schematic of plasmonic colorimetric strategy for enzyme-guided growth of $Ag^0$ on AuNS and LSPR blue-shift.

## 3.4. Glucose detection

Using optimized conditions, this assay system was investigated for the determination of various concentrations of glucose as a proof of concept. As shown in figure 6a, the LSPR shifted when the glucose concentration was increased from 1.25 to 8.75 mM. This demonstrated the strong dependence of the LSPR shift on the concentration of glucose. The inset in figure 6c shows the colours corresponding to the LSPR shifts. The colour of solutions changed from blue, in the control sample, to purple and finally to different intensities of orange. This showed the potential of the nanosensors to be applied in assays with both UV–vis spectrophometric and naked-eye readouts. The scatter plot of the glucose concentration verses the inverse of the OD max is presented in figure 6b. This suggests a predictable quantifiable relationship of the nanosensors to potentially distinguish between different concentrations of an analyte. The TEM images (figure 6c) show the morphological changes that occurred during the reaction. In the absence of glucose, the nanostructures remained star-shaped. Conversely, when glucose was added to various tubes in increasing concentrations, the AuNSs progressively became more spherical. The change in morphology (from star to spherical) could be attributed to an increase in silver coating on the surface of the nanostars, as illustrated in Scheme 1. To validate the assumption that the change in size was due to increased silver coating, two representative samples (I and III) were analysed for elemental composition. As expected, sample I consisted of Au only, while sample III contained a significant amount of silver. The analysis proved the assumption that silver was responsible for the growth or change in morphology of the AuNSs.

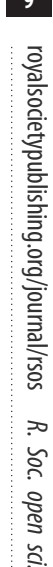

**Figure 6.** (*a*) Normalized UV–vis spectra of GOx-modified seedless-AuNSs showing blue-shift on reacting with different concentrations of glucose. (*b*) Plot of glucose concentration versus inverse maximum absorption. (*c*) TEM images of the seedless-AuNSs in the presence of increasing glucose concentrations: (I) 0 mM, (II) 1.25 mM, (III) 5 mM and (IV) 8.75 mM. (*d*) EDS analysis of C-I and III of the seedless-AuNSs.

Therefore, this demonstrated that the nanostars when properly functionalized with enzymes could be conveniently used to detect different amounts of analytes of interest.

# 4. Conclusion

A surfactant-free seedless one-pot method for AuNSs synthesis, which had advantages of both the seeded strategies and seedless strategies, was developed. These seedless AuNSs were simple and rapid to synthesize, with predictable growth of branches. The absorption peak of AuNSs was above 650 nm with a size of approximately 59 nm. This improves sensitivity in blue-shifted plasmonic LSPR. The AuNSs exhibited good stability when capped with PVP as a stabilizer. When investigated for plasmonic colorimetric sensing using glucose as a model analyte, they exhibited great stability in ionic environments and sensitivity in detection. This suggests that they are suitable transducers for biosensing applications. Further studies are being conducted in our laboratory on the optimal functionalization for further applications in plasmonic LSPR sensing in complex biological matrices.

Ethics. Research Ethics. Ethical approval to carry out this study was granted by the North-West University Research Ethics Committee.
Data accessibility. Data for this manuscript can be accessed from our university repository via this link: http://dx.doi.org/10.5061/dryad.78nb548 [27].
Authors' contributions. M.M.P. conceived the study, designed the study, carried out the synthesis, modification and glucose testing laboratory work, participated in data analysis and drafted the manuscript; B.C.V. and D.W.M. participated in design of the study, data analysis and drafting of the manuscript. All authors gave final approval for publication.
Competing interests. We have no competing interests.

Funding. All authors were supported by the North-West University's Centre of Human Metabolomics (CHM) and the South African Technology Innovation Agency (TIA) to carry out this work.

Acknowledgements. The authors thank Dr Anine Jordaan from the Laboratory for Electron Microscopy, Chemical Resource Beneficiation, North-West University, Potchefstroom, South Africa for the assistance with nanoparticle characterization and imaging.

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
