## [Reviewer comments · Royal Society Open Science]

Review History

RSOS-181971.R0 (Original submission)

Review form: Reviewer 1

Is the manuscript scientifically sound in its present form?

Yes

Are the interpretations and conclusions justified by the results?

No

Is the language acceptable?

No

Is it clear how to access all supporting data?

Not Applicable

Do you have any ethical concerns with this paper?

No

Have you any concerns about statistical analyses in this paper?

No

Recommendation?

Major revision is needed (please make suggestions in comments)

Comments to the Author(s)

Authors report seedless synthesis of gold nanostars of 60 nm size and their application in biosensing. Typically they have used ascorbic acid reduction of gold salt in acidic medium in presence of Ag ion. Advantage and appropriateness of synthetic condition need to be varified/compared with various control conditions. In particular sequence of addition of various reagents and their consequence in optical property, particle size and quality of nanostars should be studied. Glucose detection approach via formation Ag form hydrogen peroxide, is interesting. The language is too difficult to follow.

Review form: Reviewer 2

Is the manuscript scientifically sound in its present form?

Yes

Are the interpretations and conclusions justified by the results?

Yes

Is the language acceptable?

Yes

Is it clear how to access all supporting data?

Yes

Do you have any ethical concerns with this paper?

No

Have you any concerns about statistical analyses in this paper?

No

Recommendation?

Accept with minor revision (please list in comments)

Comments to the Author(s)

The authors (Phiri et al.) present an adapted technique for synthesis of gold nanostars (AuNS) to provide more control and stability than the recent HEPES method, through use of silver ions and citrate as the reductant. The nanoparticles produced are well characterised and applied in a silver deposition assay for glucose sensing.

The work seems novel, and sound and I have a only a few small comments for the authors:

1) It would be good to see the DLS intensity data curves plotted out in the paper or SI, rather than just the numbers, to get a better idea of the quality of fit and make it easier for the reader. The way the data is currently provided is hard to use.

- 2) In Figures 2 and 6, whilst the TEM is nice, the scalebars should be made clearer.
- 3) On p5 line 10, is there any sensitivity of this new method at all to the addition time of the Ag, or is the Ag the reaction initiator? (How immediate is "immediate addition of AuNS" to get good AuNS?)
- 4) Why was 0.3 M of NaCl used as the salt stability test concentration? Do the authors have a feel for that the upper limit of stability to salt and pH might be?
- 5) Typo on P7 line 45 "In the absence of glucose, the seedless AuNSs remained star shaped and progressively became more spherical as the glucose concentration was increased." This seems to conflate two different processes - absence = star shape, presence = AuNS become spherical?
- 6) Is Figure 6a normalised? If so, please state in the caption.
- 7) Suggested references to include on similar or related, rapid routes to AuNS and theory on their growth mechanisms:
 - a. 10.1021/acs.chemmater.6b03242
 - b. 10.1039/C3RA44842H
 - c. 10.1002/adma.201605622

Decision letter (RSOS-181971.R0)

18-Dec-2018

Dear Mr Phiri:

Title: Seedless gold nanostars with seed-like advantages for biosensing applications
Manuscript ID: RSOS-181971

The editor assigned to your manuscript has now received comments from reviewers. We would like you to revise your paper in accordance with the referee and Subject Editor suggestions which can be found below (not including confidential reports to the Editor). Please note this decision does not guarantee eventual acceptance.

Please submit your revised paper before 10-Jan-2019. Please note that the revision deadline will expire at 00.00am on this date. If we do not hear from you within this time then it will be assumed that the paper has been withdrawn. In exceptional circumstances, extensions may be possible if agreed with the Editorial Office in advance. We do not allow multiple rounds of revision so we urge you to make every effort to fully address all of the comments at this stage. If deemed necessary by the Editors, your manuscript will be sent back to one or more of the original reviewers for assessment. If the original reviewers are not available we may invite new reviewers.

When submitting your revised manuscript, you must respond to the comments made by the referees and upload a file "Response to Referees" in "Section 6 - File Upload". Please use this to document how you have responded to the comments, and the adjustments you have made. In

order to expedite the processing of the revised manuscript, please be as specific as possible in your response.

On behalf of the Subject Editor Professor Anthony Stace and the Associate Editor Professor Claire Carmalt.

RSC Associate Editor:
Comments to the Author:
(There are no comments.)

RSC Subject Editor:
Comments to the Author:
(There are no comments.)

Reviewers' Comments to Author:
Reviewer: 1

Comments to the Author(s)

Authors report seedless synthesis of gold nanostars of 60 nm size and their application in biosensing. Typically they have used ascorbic acid reduction of gold salt in acidic medium in presence of Ag ion. Advantage and appropriateness of synthetic condition need to be varified/compared with various control conditions. In particular sequence of addition of various reagents and their consequence in optical property, particle size and quality of nanostars should be studied. Glucose detection approach via formation Ag form hydrogen peroxide, is interesting. The language is too difficult to follow.

Reviewer: 2

Comments to the Author(s)

The authors (Phiri et al.) present an adapted technique for synthesis of gold nanostars (AuNS) to provide more control and stability than the recent HEPES method, through use of silver ions and citrate as the reductant. The nanoparticles produced are well characterised and applied in a silver deposition assay for glucose sensing.

The work seems novel, and sound and I have a only a few small comments for the authors:

- 1) It would be good to see the DLS intensity data curves plotted out in the paper or SI, rather than just the numbers, to get a better idea of the quality of fit and make it easier for the reader. The way the data is currently provided is hard to use.
- 2) In Figures 2 and 6, whilst the TEM is nice, the scalebars should be made clearer.
- 3) On p5 line 10, is there any sensitivity of this new method at all to the addition time of the Ag, or is the Ag the reaction initiator? (How immediate is “immediate addition of AuNS” to get good AuNS?)
- 4) Why was 0.3 M of NaCl used as the salt stability test concentration? Do the authors have a feel for that the upper limit of stability to salt and pH might be?
- 5) Typo on P7 line 45 “In the absence of glucose, the seedless AuNSs remained star shaped and progressively became more spherical as the glucose concentration was increased.” This seems to conflate two different processes - absence = star shape, presence = AuNS become spherical?
- 6) Is Figure 6a normalised? If so, please state in the caption.
- 7) Suggested references to include on similar or related, rapid routes to AuNS and theory on their growth mechanisms:
 - a. [10.1021/acs.chemmater.6b03242](https://doi.org/10.1021/acs.chemmater.6b03242)
 - b. [10.1039/C3RA44842H](https://doi.org/10.1039/C3RA44842H)
 - c. [10.1002/adma](https://doi.org/10.1002/adma).

Author's Response to Decision Letter for (RSOS-181971.R0)

See Appendix A.

Decision letter (RSOS-181971.R1)

14-Jan-2019

Dear Mr Phiri:

Title: Seedless gold nanostars with seed-like advantages for biosensing applications
Manuscript ID: RSOS-181971.R1

It is a pleasure to accept your manuscript in its current form for publication in Royal Society Open Science. The chemistry content of Royal Society Open Science is published in collaboration with the Royal Society of Chemistry.

On behalf of the Subject Editor Professor Anthony Stace and the Associate Editor Professor Claire Carmalt.

RSC Associate Editor
Comments to the Author:
(There are no comments.)

Reviewer(s)' Comments to Author:

Appendix A

Response to Referees

Referee 1

- Advantage and appropriateness of synthetic condition need to be verified/compared with various control conditions. In particular sequence of addition of various reagents and their consequence in optical property, particle size and quality of nanostars should be studied.

Response: *It was experimentally observed that even when HCl was not added in the synthesis procedure, gold nanostars were still formed. However, the localised surfaced plasmon resonance of these nanostars were more blue-shifted compared to those synthesised with HCl added to it. This corresponds to what Yuan et al also discovered in their investigation of the effect of pH on the nanostars formation [1]. They further investigated the effect of lowering the pH by substituting HCl with nitric acid. They observed that the nanostars were not formed at all. This led to the postulation that the Cl⁻ ions could be involved in the formation of the gold nanostars. The effects of addition or omission of HCl on the size of the nanostars were not studied in this manuscript. Other studies using the ascorbic acid silver-assisted nanostars recipes adequately report the experimental effects of the various combinations of ascorbic acid in relation to the HAuCl₄, as well as adding varying quantities of silver nitrate to the mixture [1-3]. Thus, we opted to mostly focus on the removal of the gold seeds and strategically changing the order of the addition of the reagents.*

- Glucose detection approach via formation Ag form hydrogen peroxide, is interesting. The language is too difficult to follow.

Response: *The language here was revised and simplified so as to make it easy to follow. It now reads as follows as quoted exactly from the manuscript:*

Using optimized conditions, this assay system was investigated for the determination of various concentrations of glucose as a proof of concept. As shown in Figure 6a, the LSPR shifted when the glucose concentration was increased from 1.25 to 8.75 mM. This demonstrated the strong dependence of the LSPR shift on the concentration of glucose. The insert in Figure 6c shows the colors corresponding to the LSPR shifts. The solutions changed color from blue in the control sample, to purple and finally to different intensities of orange. This showed the potential of the nanosensors to be applied in assays with both UV-vis spectrophotometric- and naked-eye readouts. The scatter plot of the glucose concentration verses the inverse of the OD max is presented in Figure 6b. This suggests a predicable quantifiable relationship of the nanosensors to potentially distinguish between different concentrations of an analyte. The TEM images (Figure 6c) show the morphological changes that occurred during the reaction. In the absence of glucose, the nanostructures remained

star-shaped. Conversely, when the glucose was added to various tubes in increasing concentrations, the AuNS progressively became more spherical in shape. The change in morphology from star to spherical could be attributed to an increase in silver coating on the surface of the nanostars, as illustrated in Scheme 1. To validate the assumption that the change in size was due to increased silver coating, two representative samples (I and III) were analyzed for elemental composition. As expected, sample I consisted of Au only while sample III contained a significant amount of silver. The analysis proved the assumption that silver was responsible for the growth or change in morphology of the AuNSs. Therefore, this demonstrated that the nanostars when properly functionalized with enzymes could be conveniently used to detect different amounts of analytes of interest.

Referee 2

- 1) It would be good to see the DLS intensity data curves plotted out in the paper or SI, rather than just the numbers, to get a better idea of the quality of fit and make it easier for the reader. The way the data is currently provided is hard to use.

This was done and now added to Figure 3.

- 2) In Figures 2 and 6, whilst the TEM is nice, the scalebars should be made clearer.

The images were reworked and the scalebars are now clearer.

- 3) On p5 line 10, is there any sensitivity of this new method at all to the addition time of the Ag, or is the Ag the reaction initiator? (How immediate is “immediate addition of AuNS” to get good AuNS?)

The time for the addition of AgNO_3 was observed to be very important in this method. In the experimental protocol, AgNO_3 was added about 10 seconds after the addition of HAuCl_4 . This led to star formation. On the other hand, prolonged delay of over 30 seconds in the addition of AgNO_3 led to the formation of other nanoshapes with a distinctive pink/light purple coloured solution.

- 4) Why was 0.3 M of NaCl used as the salt stability test concentration? Do the authors have a feel for that the upper limit of stability to salt and pH might be?

Application of protein-modified gold nanostars in physiological pH and ionic environment requires that they are stable in solutions containing high concentration of proteins and salts. Thus, the stability of the GOx-modified seedless-AuNSs was tested and proved in 0.3 M NaCl and pH 7 [4, 5]. 0.3 M NaCl is deemed to be twice the ionic concentration of salt in plasma which is 0.135 M [6]. Therefore, this concentration has been in the cited publication to test the

stability of the nanoparticles in ionic environment [4]. A critical flocculation concentration is usually determined as the threshold concentration of NaCl in the gold solution which causes rapid aggregation [5]. However, this was not done for the submitted manuscript.

- 5) Typo on P7 line 45 “In the absence of glucose, the seedless AuNSs remained star shaped and progressively became more spherical as the glucose concentration was increased.” This seems to conflate two different processes – absence = star shape, presence = AuNS become spherical?

The line was revised and now reads as follows:

“In the absence of glucose, the nanostructures remained star-shaped. Conversely, when the glucose was added to various tubes in increasing concentrations, the AuNS progressively became more spherical in shape.”

- 6) Is Figure 6a normalised? If so, please state in the caption.

Yes. Figure 6a was normalized and has since been stated in the caption.

- 7) Suggested references to include on similar or related, rapid routes to AuNS and theory on their growth mechanisms:

- a. 10.1021/acs.chemmater.6b03242
- b. 10.1039/C3RA44842H
- c. 10.1002/adma.201605622

The suggested references were reviewed and appropriately included in the manuscript. The suggestions are much appreciated as they threw some light on the subject matter.

Bibliography

- 1 Yuan, H., Khoury, C. G., Hwang, H., Wilson, C. M., Grant, G. A., Vo-Dinh, T. 2012 Gold nanostars: surfactant-free synthesis, 3D modelling, and two-photon photoluminescence imaging. *Nanotechnology*. **23**, 075102.
- 2 Ahmed, W., Kooij, E. S., Van Silfhout, A., Poelsema, B. 2010 Controlling the morphology of multi-branched gold nanoparticles. *Nanotechnology*. **21**, 125605.
- 3 Kawamura, G., Yang, Y., Fukuda, K., Nogami, M. 2009 Shape control synthesis of multi-branched gold nanoparticles. *Materials Chemistry and Physics*. **115**, 229-234.
- 4 Rodríguez-Lorenzo, L., De La Rica, R., Álvarez-Puebla, R. A., Liz-Marzán, L. M., Stevens, M. M. 2012 Plasmonic nanosensors with inverse sensitivity by means of enzyme-guided crystal growth. *Nature materials*. **11**, 604.
- 5 Wangoo, N., Bhasin, K. K., Mehta, S. K., Suri, C. R. 2008 Synthesis and capping of water-dispersed gold nanoparticles by an amino acid: bioconjugation and binding studies. *J Colloid Interface Sci*. **323**, 247-254. (10.1016/j.jcis.2008.04.043)
- 6 Marshall, W. J., S. K. Bangert, and Marta Lapsley. 2008 *Clinical Chemistry*. Mosby Elsevier.